# Disentangling Information in Artificial Images of Plant Seedlings Using Semi-Supervised GAN

**Simon Leminen Madsen [1,*]** , **Anders Krogh Mortensen [2]** , **Rasmus Nyholm Jørgensen [1]** and **Henrik Karstoft [1]**

1  Department of Engineering, Aarhus University, Finlandsgade 22, DK-8200 Aarhus N, Denmark;
   rnj@eng.au.dk (R.N.J.); hka@eng.au.dk (H.K.)
2  Department of Agroecology, Aarhus University, Forsøgsvej 1, DK-4200 Slagelse, Denmark;
   anmo@agro.au.dk
*  Correspondence: slm@eng.au.dk

**Abstract:** Lack of annotated data for training of deep learning systems is a challenge for many visual recognition tasks. This is especially true for domain-specific applications, such as plant detection and recognition, where the annotation process can be both time-consuming and error-prone. Generative models can be used to alleviate this issue by producing artificial data that mimic properties of real data. This work presents a semi-supervised generative adversarial network (GAN) model to produce artificial samples of plant seedlings. By applying the semi-supervised approach, we are able to produce visually distinct samples for nine unique plant species using a single GAN model, while still maintaining a relatively high visual variance in the produced samples for each species. Additionally, we are able to control the appearance of the generated samples with respect to rotation and size through a set of latent variables, despite these not being annotated features in the training data. The generated samples resemble the intended species with an average recognition accuracy of ∼64.3%, evaluated using an external state-of-the-art plant seedling classification model. Additionally, we explore the potential of using the GAN model's discriminator as a quality assessment tool to remove poor representations of plant seedlings from the artificial samples.

**Keywords:** generative model; generative adversarial networks; supervised learning; unsupervised learning

## 1. Introduction

Machine learning, and in particular deep learning, have become increasingly popular in recent years. However, deep learning algorithms are still primarily trained using supervised learning, which requires a substantial amount of annotated training data. There exist many publicly available datasets for training deep learning systems [1]; however, when it comes to domain-specific applications such as plant detection and recognition, the availability of data is significantly reduced [2].

Alternatively, generative models can be used to extend the amount of available data by producing artificial data samples that mimic properties of real data. Recently, a new type of generative models called generative adversarial networks (GANs) were introduced by Goodfellow et al. [3]. While GAN models can be used in many different generative applications, they are most often used for generating artificial images. A strength of the GAN models is that they can be trained using an unsupervised and/or supervised end-to-end learning approach [4–6]. Previous research has shown that artificial GAN samples can be used as data augmentation to increase the performance of several visual recognition tasks [7–10].

GAN models have also been applied to the modeling of plants or plant seedlings. The primary focus has been on increasing the amount of available training data for applications, such as leaf counting [11,12] or species classification [13]. The aforementioned models all use a supervised learning approach, conditioning the models based on leaf count, plant species, or full segmentation masks. Besides this conditioning, the models provide little control over the appearance of the generated samples. In other domains, unsupervised learning schemes have been used to adopt greater control of over the appearance of the generated samples. For example, the infoGAN model by Chen et al. [4] has shown that it is possible to adapt dominating modalities in the data by adding additional unsupervised conditioning on the model.

In this work, we present an extension of the WacGAN model by [13], which is used to produce artificial images of plant seedlings. We designate this extended model WacGAN-info. The extension consists of adding an unsupervised learning branch to the GAN configuration, similar to infoGAN [4], which enables control over the artificial samples' visual appearance through an additional set of latent input variables. The unsupervised learning branch also enables the WacGAN-info model to produce samples that visually and quantitatively have a greater resemblance to real plant seedlings, compared with those produced by the WacGAN model. Additionally, we show that the GAN disciminator network can be used as a tool to prevalidate the quality of the artificial WacGAN-info samples. This prevalidation enables us to remove samples with poor class representation from the artificial data distribution, which leads to a further increase in resemblance to real samples.

*Related Work*

GANs have been widely studied in recent years, and several improvements have been proposed to accommodate issues such as poor convergence properties and mode collapse [5,14].

The WGAN model by Arjovsky et al. [15] proposed an alternative objective formulation based on the Wasserstein distance to improve the convergence properties and sample quality of the GAN model. The WGAN-GP model by Gulrajani et al. [16] further improved this objective formulation of the Wasserstein objective by introducing a regularization term to avoid weight clipping of the model's network weights.

Conditioning on the GAN model can be used to enable better control of the visual content in the generated samples. Conditions can be applied supervised, unsupervised, or in a mixture of the two. Additionally, conditioning also ensures better convergence of the GAN [5]; e.g., the infoGAN model by Chen et al. [4] applies an unsupervised conditioning approach to adapt dominating modalities in the training data. To achieve this, the infoGAN model is trained with an additional classification objective that ensures information preservation throughout the model. This allows Chen et al. [4] to control the content of the generated samples through a set of latent info variables. The ACGAN model by Gulrajani et al. [16] applies a supervised conditioning approach to produces distinguishable samples of all 1000 ImageNet classes [17]. The approach is similar to that of infoGAN [4]; however, in the ACGAN model, the latent variables are directly linked to the class labels of the training data. There exist more advanced conditioning schemes, e.g., conditioning on full images [18–20] or text encodings [21,22]; however, these conditioning schemes require a high level of detail in the annotations.

Valerio Giuffrida et al. [12] and Zhu et al. [11] both use supervised conditioning to train a GAN model to produce artificial samples of *arabidopsis* plants. The artificial GAN samples were in both cases used for data augmentation and thus increased the performance on leaf counting. Valerio Giuffrida et al. [12] use conditioning based directly on the leaf count, whereas Zhu et al. [11] use full segmentation masks for conditioning. However, both these models are limited to producing samples from a single species. Madsen et al. [13] applied a single GAN model to generate artificial image samples of multiple plant seedling species. This model combined results from WGAN-GP [16] and ACGAN [6] to produce distinguishable samples for each species while maintaining relatively high visual variance within each of these. Additionally, Madsen et al. [13] applied the artificial samples in a

transfer-learning setup for plant seedling classification, where they provided a strong basis for further fine-tuning.

## 2. Methods

GANs are a configuration of two neural network architectures: a generator network, $G$, and a discriminator network, $D$. Both networks are trained using an adversarial learning scheme, where the two networks are competing against each other. $D$ is responsible for distinguishing between the samples from the real and artificial data distributions, whereas $G$ produces artificial samples that mimic real samples in order to cheat $D$ into making mistakes.

The GAN model presented in this work is an extension of the WacGAN model by Madsen et al. [13]. The extension incorporates an additional unsupervised learning branch, as inspired by the infoGAN model by Chen et al. [4]. We designate this configuration WacGAN-info. WacGAN-info is a semi-supervised generative model that is capable of producing visually distinct samples for each class while still maintaining a relatively high amount of variance within each of these. The model also provides control of the visual content through a set of latent variables. This is achieved by adding a noise variable to the generator input, which the discriminator output is forced to recreate from the generated sample.

The configuration of WacGAN-info is visualized in Figure 1. The figure shows how $G$ inputs three vectors: $z$, $y$, and $u$, representing random noise, class encoding, and latent info encoding, respectively. The output from $G$ is an artificial data sample, $\tilde{x}$. Additionally, $D$ inputs either real data samples, $x$, or artificial data samples, $\tilde{x}$. The output of $D$ are three parallel signals: $D_S$, $D_C$, and $D_I$, which correspond to the estimation of the data source distribution (real or artificial samples), the class, and the latent info variable, respectively.

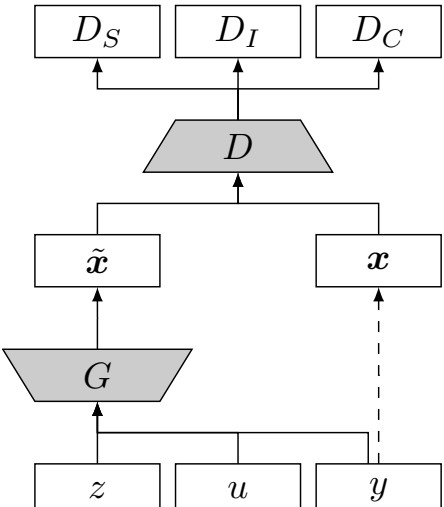

**Figure 1.** WacGAN-info model configuration. $D$ is the discriminator network, and $G$ is the generator network. $z$, $y$, and $u$ are the generator inputs corresponding to noise vector, class vector, and latent info vector, respectively. The signals $x$ and $\tilde{x}$ are real and artificial data samples. Finally, $D_S$, $D_C$, and $D_I$ are the parallel discriminator outputs corresponding to the source distribution estimation (real or artificial samples), the class estimation, and the info variable estimation.

### 2.1. Objective Function

The objective function is divided into three parts: a source loss, $L_S$, a class loss, $L_C$, and an info loss, $L_I$.

The source loss, $L_S$, is responsible for distinguishing between the real and artificial data distributions given the discriminator output, $D_S$. $L_S$ is implemented as a Wasserstein loss with

gradient penalty, which provides a high degree of diversity in the generated samples [16]. $L_S$ is given by

$$L_S = \mathop{\mathbb{E}}_{x \sim p_r} [D_S(x)] - \mathop{\mathbb{E}}_{\tilde{x} \sim p_g} [D_S(\tilde{x})] + \lambda \mathop{\mathbb{E}}_{\hat{x} \sim p_{\hat{x}}} \left[ (\|\nabla_{\hat{x}} D_S(\hat{x})\|_2 - 1)^2 \right], \quad (1)$$

where $p_r$ is the real data distribution and $p_g$ is the artificial data distribution implicitly defined by $\tilde{x} = G(z, \tilde{y}, u)$, $z \sim p(z)$, [16]. $p_{\hat{x}}$ is a random sample distribution generated from random interpolation between the real and artificial samples [16].

The class loss, $L_C$, is responsible for guiding the generator to produce samples, which are visually distinguishable for each separate class. $L_C$ is implemented as the cross-entropy loss between the one-hot encoded class label, $y$, and the output of the discriminator's classification branch, $D_C(x)$ [4,23]. $L_C$ is given by:

$$L_C = \mathop{\mathbb{E}}_{x \sim p_r} \left[ - \sum_{k \in N_C} y_k \log D_{C_k}(x) \right] + \mathop{\mathbb{E}}_{\tilde{x} \sim p_g} \left[ - \sum_{k \in N_C} \tilde{y}_k \log D_{C_k}(\tilde{x}) \right], \quad (2)$$

where $N_C$ is the number of classes.

Finally, the info loss, $L_I$, ensures information preservation throughout the network, thus allowing the content of the generated samples to be controlled through a set of latent variables, $u$, as inspired by Chen et al. [4]. $L_I$ is implemented as the mean square loss between the intended info variables $u$ and the predicted info variables of the discriminator $D_I(x)$ and is given by

$$L_I = \mathop{\mathbb{E}}_{\tilde{x} \sim p_g} \left[ \sum_{i \in N_I} (u_i - D_{I_i}(\tilde{x}))^2 \right], \quad (3)$$

where $N_I$ is the number of latent variables.

The discriminator is trained to minimize $-L_S + w_C L_C + w_I L_I$, whereas the generator is trained to minimize $L_S + w_C L_C + w_I L_I$. $w_C$ and $w_I$ are weight terms used to regulate the relative importance between the source loss, the class loss, and the info loss. The values of the $w_C$ and $w_I$ are determined empirically through experiments.

*2.2. Network Design*

The network architectures of the WacGAN-info model are inspired by the ACGAN model by [6]. As WacGAN-info is an extension of previous work [13], we apply the same modifications of the original ACGAN architecture. Additionally, the networks have been modified to accommodate for the additional information branch by adding a set of latent variables, $u$, as input to the generator and an additional output, $D_I$, to the discriminator to reproduce $u$. $u$ are drawn from a random uniform distribution $p(u) \sim U(-1, 1)$. $u$ are concatenated with the categorical class encoding $y$ and the noise vector $z$. The $D_I$ is implemented as a fully connected layer in parallel with the source discriminator, $D_S$, and the class discriminator, $D_C$, (Figure 1). The full implementation is summarized in Appendix A.1.

*2.3. Evaluation*

The artificial samples were evaluated qualitatively and quantitatively to assess their quality. In the qualitative evaluation, we performed a visual inspection of the artificial samples to provide a subjective assessment of the sample's quality. The visual inspection was primarily used to assess the realism of the artificial samples (Does the sample look as a realistic plant?), as to our knowledge, there does not exist a quantitative metric to measure the realism of images.

In the quantitative evaluation, the class discriminability (How well does the artificial sample represent the intended species?) was evaluated by processing each sample using an external state-of-the-art classifier trained for plant species classification. Previously, the inception score [24] has been widely adopted as a metric for measuring the class discriminability in GAN research [5,6,16,21]. Madsen et al. [13] apply a similar approach where the external classifier is instead based on the

ResNet-101 architecture [25], which is fine-tuned for plant seedling classification. To be able to directly compare the results, we applied the same external classifier as in [13]. The configuration of the external ResNet-101 classifier is summarized in Appendix A.2. The class discriminability is reported as the recognition accuracy, which indicates how often the artificial samples are correctly classified as the intended species by the external classifier [13]. Perfectly generated artificial samples will be indistinguishable from the real samples in terms of classification accuracy by the ResNet-101 classifier.

We applied the same leave-$p$-out cross validation scheme as Madsen et al. [13] to avoid data leakage in the quantitative results [26] and to be able to directly compare our results to Madsen et al. [13]. By splitting the real data into four parts and using a $p$ of 50%, we get six combinations of training and test sets. For each combination, the external ResNet-101 classifier was trained on the associated training set and the WacGAN-info model was trained on the corresponding test set. Thus, the WacGAN-info model could be used to generate an additional test-set of artificial samples for the class discriminability test on the ResNet-101 classifier.

## 3. Results

Both the WacGAN-info model and the external ResNet-101 classifier were trained on the segmented plant seedling dataset (sPSD) by Giselsson et al. [27]. This dataset consists of segmented RBG images of plant seedlings from twelve different species. However, in this work, we have ignored all grass species as these are not properly instance segmented. This left nine species ($N_C = 9$) remaining in the dataset. The plant species are identified by their European and Mediterranean Plant Protection Organization (EPPO) codes [28] throughput the paper. Due to the network's architecture, the WacGAN-info model only supports input images with a resolution of $128 \times 128$ pixels. To maintain a relatively low resizing factor, all images with a resolution of $>400$ pixels in either dimension are excluded from the dataset. This results in a resizing factor of $400/128 = 3.125$. The remaining species and number of samples for each of these are summarized in Table 1.

**Table 1.** Overview of data in sPSD after removing grass species and images with a resolution larger than 400 pixels in either dimension.

| Common Name | Latin Name | EPPO Code | N Samples |
|---|---|---|---|
| Charlock | *Sinapis arvensis* | SINAR | 297 |
| Cleavers | *Galium aparine* | GALAP | 270 |
| Common chickweed | *Stellaria media* | STEME | 591 |
| Fat hen | *Chenopodium album* | CHEAL | 447 |
| Maize | *Zea mays* | ZEAMX | 149 |
| Scentless mayweed | *Matricaria perforata* | MATIN | 498 |
| Shepherd's purse | *Capsella bursa-pastoris* | CAPBP | 225 |
| Small-flowered cranesbill | *Geranium pusillum* | GERPU | 429 |
| Sugar beets | *Beta vulgaris* var. *altissima* | BEAVA | 191 |

To train the WacGAN-info model, all of the images were preprocessed by: zero-padding to achieve a resolution of $470 \times 470$ pixels, random rotation of the image, central crop of $400 \times 400$ pixels, and finally, resizing to $128 \times 128$ pixels using bilinear interpolation. This data augmentation approach is also used for training of the external ResNet-101 classifiers.

The regularization weights, $w_C$ and $w_I$, are set to 7.5 and 15, respectively. As mentioned, these values are determined empirically and provide a good trade-off between intra- and inter-class diversity in the artificial samples, while allowing the latent variables to adapt dominating modalities in the data. The length of the noise vector, $z$, is set to 128, as inspired by previous work [13]. The following results are produced using two latent variables ($N_I = 2$). A full summary of the hyperparameters can be found in Appendix A.1.

*3.1. Visual Image Quality Assessment*

Table 2 show examples of artificial plant seedlings generated using the WacGAN-info model. The table also includes examples of real plant seedlings and artificial samples from the WacGAN model to provide references for the visual evaluation. The 16 WacGAN-info samples for each species were generated using the same fixed random noise vectors—only the class encoding changes. This shows that the WacGAN-info model is capable of producing visually distinct samples for each of the different species present in the sPSD, as the appearance of the samples changes for each column.

**Table 2.** Examples of plant seedlings from sPSD [27], WacGAN [13], and WacGAN-info. The 16 samples for each species were generated using the same fixed random noise vectors, with only the class encoding changing. The background for all samples has been changed from black to white to increase the contrast.

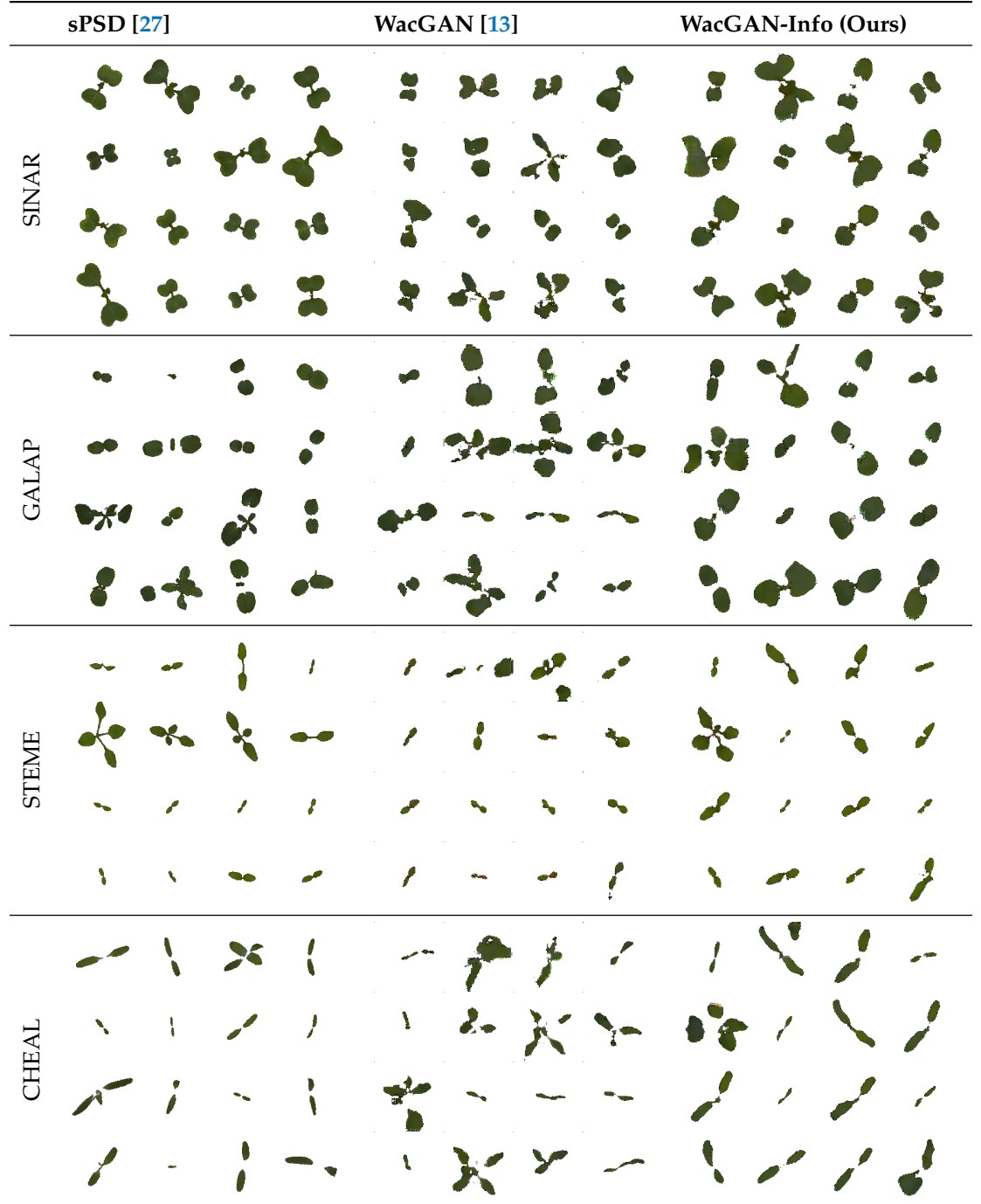

**Table 2.** *Cont.*

| sPSD [27] | WacGAN [13] | WacGAN-Info (Ours) |
| --- | --- | --- |

The artificial samples themselves are composed of a number of leaves that are distributed around the image/plant center. Compared to the samples produced in Madsen et al. [13], the WacGAN-info model is capable of generating more detailed shape and textural features; e.g., the WacGAN-info model has become better at generating thin features such as stems or small leaves. The texture across a single leaf can also assume different gradients of green, whereas previously, the texture mainly consisted of a single green color. Additionally, the borders between the plant and background are more smooth for the WacGAN-info sample, whereas the WacGAN samples [13] are more rough around the edges.

At closer inspection of the generated samples, some image artefacts can be observed. The artefacts can be described as pixel errors, which mainly occur along the border of the artificial samples or in a systematic grid in the texture. Figure 2 shows a severe example of these errors.

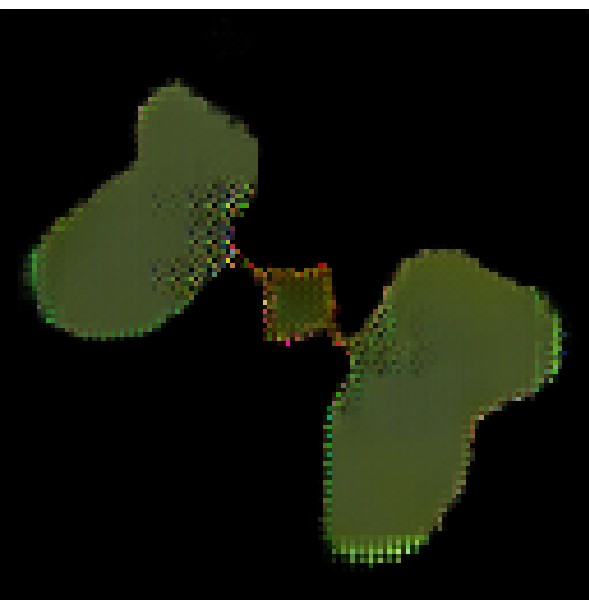

**Figure 2.** Example of image artefacts in an artificially generated SINAR sample.

### 3.1.1. Disentangled Information Space

Figure 3 shows how the appearance of the artificial samples depends on the values of the two latent variables, $u_1$ and $u_2$. Although a few outliers are observed, the latent variables can, in this example, be used to control the rotation and the size of the artificial plant samples; e.g., the samples in the left-hand side of the plot are larger compared to those in the right-hand side, indicating that the size of the artificial samples is inversely related to the value of $u_1$. Additionally, the samples in the bottom of the plot are primarily oriented vertically with respect to the primary axis of the plant, whereas samples in the top are oriented horizontally, indicating that $u_2$ models the rotation of the samples.

Figure 3 only show the results from a single species, but a similar trend is observed across the other species as well. It should be noted that this specific trend is not unanimous across the six different models as they are trained on different parts of sPSD.

### 3.2. Class Discriminability

To assess the class discriminability of the artificial samples, 10,000 samples from each class ere generated from each of the six WacGAN-info generators and subsequently evaluated on the corresponding external ResNet101 classifiers along with the corresponding real data test sets (Table 3). The artificial data achieves 64.63% average recognition accuracy when analyzed on the external classifiers. The average accuracy for recognizing real data samples is of course better compared to the artificial. However, the best-performing class on the artificial data (STEME) performs better than the worst-performing class on the real data (CAPBP). The average recognition accuracy for each class

shows a strong positive linear correlation to the number of real samples for each class in both the real data ($\rho = 0.70$) and the artificial data ($\rho = 0.85$).

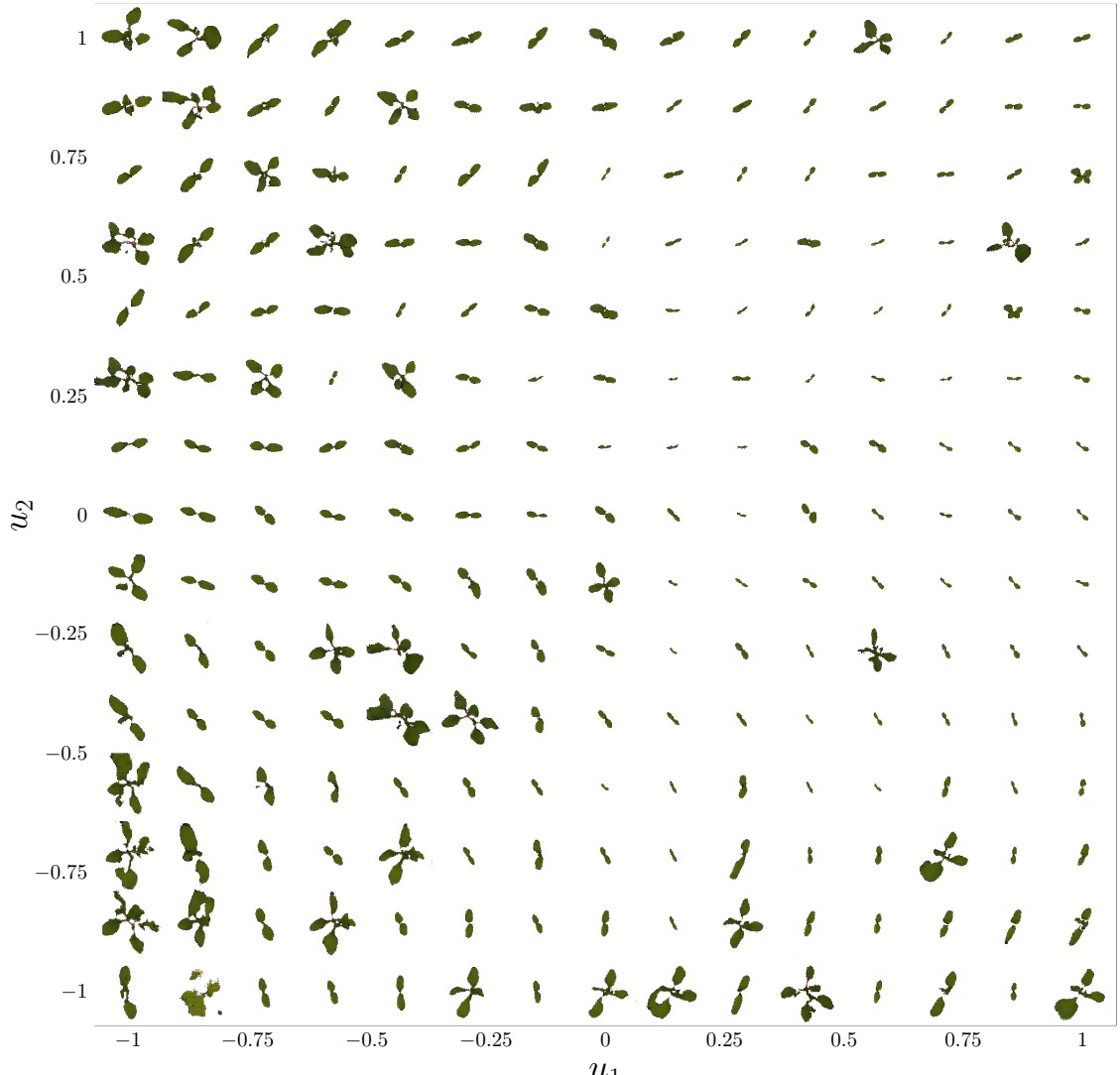

**Figure 3.** Visualization of the information space using two uniformly distributed continuous variables. The visualization shows that specific combinations of $u_1$ and $u_2$ can be used to control the rotation and size of the generated plant seedlings. All samples presented in the plot are supposed to represent STEME.

From the confusion matrix (Figure 4), it is clear that there is a good correspondence between the true labels and the predicted labels on the artificially generated data. The misclassifications (the off-diagonal entries in the confusion matrix) are generally fairly uniformly distributed among the other species; however, a few notable misclassifications do appear, which are significantly larger than the other misclassifications. These can be grouped into two groups: (1) one-way misclassifications and (2) two-way misclassifications. The two-way misclassifications are misclassifications between two species, which are both significantly larger than the other misclassifications (CHEAL $\rightleftarrows$ BEAVA, and MATIN $\rightleftarrows$ CAPBP). The misclassifications between CHEAL and BEAVA occur in samples mimicking the early growth stages, where both species are highly similar with respect to visual appearance. A large portion of the mislcassifications from BEAVA to CHEAL also includes examples of large broad leafs mixed with more elongated leafs, traits that occur in the real examples of CHEAL but not BEAVA. In the one-way misclassifications, there is only a significant misclassification

from one species to the other, but not the other way around (CAPBP → {STEME, GERPU, CHEAL}, ZEAMX → {STEME, MATIN, SINAR}, GALAP → SINAR, and BEAVA → MATIN). These misclassifications generally happen from species with relatively few samples to species with relatively many classes in the real dataset. A notable exception to this is the misclassification of GALAP as SINAR, which contain approximately the same number of samples in the real dataset. These misclassifications consists of two medium-sized round leafs connected by a small stem. Although the GALAP real data does contain samples with these leafs, the trait occurs much more frequently in the SINAR real data.

**Table 3.** The accuracy of recognizing the intended species of the artificial samples from WacGAN-info, when analyzed using an external ResNet101 classifier trained solely on real data samples from sPSD. The recognition accuracies are reported as the mean and standard deviation over the six combinations in the cross-validation scheme.

| Species | Test Set | | | |
| --- | --- | --- | --- | --- |
| | **Real Data** | | **Artificial Data** | |
| | **N Samples** | **Accuracy** | **N Samples** | **Accuracy** |
| SINAR | 149 | $94.3 \pm 1.1\%$ | 10,000 | $61.7 \pm 21.8\%$ |
| GALAP | 135 | $93.5 \pm 2.3\%$ | 10,000 | $63.6 \pm 10.4\%$ |
| STEME | 296 | $96.3 \pm 0.7\%$ | 10,000 | $89.0 \pm 5.4\%$ |
| CHEAL | 224 | $95.3 \pm 1.7\%$ | 10,000 | $63.4 \pm 14.2\%$ |
| ZEAMX | 75 | $90.5 \pm 3.1\%$ | 10,000 | $47.2 \pm 16.6\%$ |
| MATIN | 249 | $94.3 \pm 1.6\%$ | 10,000 | $67.0 \pm 17.3\%$ |
| CAPBP | 113 | $84.0 \pm 3.1\%$ | 10,000 | $47.4 \pm 8.4\%$ |
| GERPU | 215 | $95.8 \pm 1.5\%$ | 10,000 | $80.7 \pm 6.5\%$ |
| BEAVA | 96 | $90.6 \pm 4.8\%$ | 10,000 | $59.0 \pm 12.0\%$ |
| Total | 1549 | $92.7 \pm 1.0\%$ | 90,000 | $64.3 \pm 4.4\%$ |

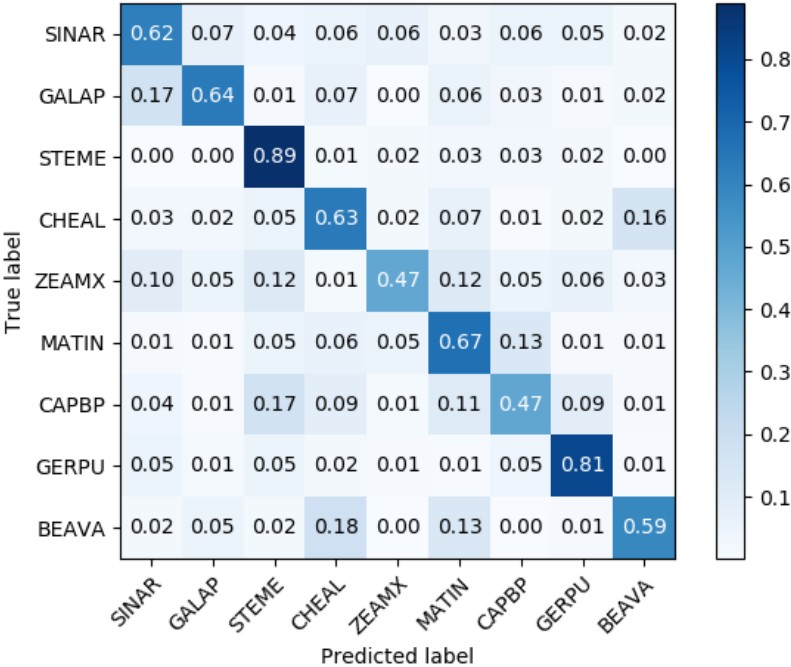

**Figure 4.** Average confusion matrix when evaluating the artificial samples from the WacGAN-info model using the external classifier trained solely on real samples. Each row is normalized with respect to the number of true labels of that class.

Using the GAN Descriminator to Prevalidate Sample Quality

The recognition accuracies reported in Table 3 are based on all samples generated by the WacGAN-info model. However, the classification branch of the discriminator, $D_C(x)$, can be used to perform a prevalidation of the artificial samples. Due to the softmax activation, $D_C(x)$ outputs a confidence of how well a sample represents each class. Thus, by examining the discriminator's confidence in the intended class for each artificial sample, we can assure better class representations. Figure 5 shows the average recognition accuracy (Figure 5a) and the sample distribution (Figure 5b) as a function of the discriminator's confidence in the samples representing the intended class. The figure shows that the artificial samples generally get recognized as the intended class more often by the external classifier, if the discriminator also has high confidence in the samples belonging to the intended class. Thus, it is possible to remove samples with poor class representations by applying a threshold on the discriminator's confidence in each sample. Additionally, Figure 5b shows that the WacGAN-info discriminator has high confidence in the majority of the artificial samples, so if we threshold on the discriminator confidence, we only remove a small part of the data; e.g., by including only samples with a discriminator confidence greater than 0.95, the average recognition accuracy across the six models increases from 64.3% to 65.3%, while still including the 86% of the artificial samples.

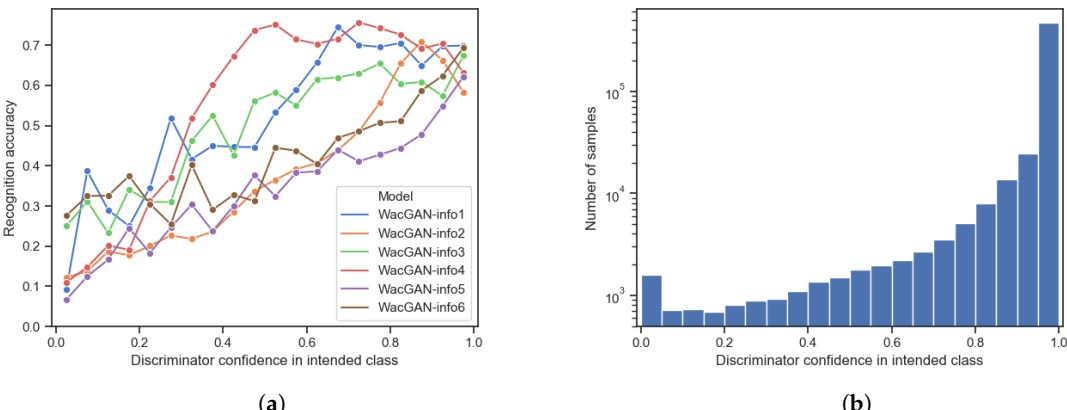

(**a**)                                                                 (**b**)

**Figure 5.** (**a**) Classifier recognition accuracy as a function of the WacGAN-info discriminator confidence in the samples belonging to the intended class, $D_C$. The artificial GAN samples are divided into 20 bins corresponding to the discriminator confidence in each. The bins are uniformly spaced in the interval $[0;1]$. From the samples in each bin, the average class recognition accuracy is evaluated using the external classifier. Each color represents one of the six models trained for cross-validation, and the markers represent the bin centers. (**b**) The histogram showing the sample distribution as a function of the WacGAN-info discriminator confidence in the samples belonging to the intended class, $D_C$.

## 4. Discussion

The results show that the WacGAN-info model is capable of producing artificial samples that resemble real plant seedlings. Furthermore, it seems that the model is capable of reproducing samples that are relatively distinguishable across the nine different plant species while simultaneously ensuring a relatively high degree of variability in the samples for each class.

Although the WacGAN-info samples visually resemble real plant seedlings, there are still some obvious errors in the artificial samples. The errors are mainly related to unrealistic or inaccurate reproductions of the plant leaves, where multiple leaves have merged into one or one has been split into two or more; e.g., inspecting the samples of ZEAMX, some of these resemble dicots although ZEAMX is a monocot species (see the first and last samples in Table 2). These visual errors may be reduced by increasing the class loss weight, $w_C$, which seems to provide better inter-class diversity but comes at the cost of reducing the intra-class diversity.

The observed pixel errors in the samples are probably related to the applied generator architecture as the border errors could be related to the relatively large output kernel ($[8 \times 8]$) and the grid pattern

matches with strides of 2 used for the up-sampling. Similar errors were observed in the original ACGAN [6] paper and in the previous work by Madsen et al. [13]; however, based on visual inspection, the issue seems to have been reduced for the WacGAN-info model.

Section 3.1.1 shows that it is possible to control the appearance of the generated samples through the latent information variables. As expected, the information variables capture dominating modalities in the data, such as the size and rotation of the samples. Although the two latent variables control similar properties in all six WacGAN-info models, it should be noted that the specific trend observed in Figure 3 is not representative of all models. This is an effect of the models being trained on different dataset splits and due to the unsupervised learning scheme applied in the information learning branch, which causes the models to adopt slightly different trends. In this work, we only present results using two uniformly distributed information variables ($N_I = 2$); the setup can easily be extended to facilitate more, but adding more latent variables quickly increases the complexity of interpreting their underlying representation.

The class discriminability test (Section 3.2) also shows that the artificial samples resemble the intended species to a relatively high degree. The reported average recognition accuracy is 64.3% (see Table 3), which is well above uniformly distributed random guessing, which would yield 11.1%. Compared to Madsen et al. [13], this result shows a 5.4% point increase in the overall recognition accuracy and a 4.4% point decrease in the standard deviation thereof. As this work is an extension of Madsen et al. [13], the GAN setup is relatively similar. tThe main difference is that the WacGAN-info model includes the additional latent information variables $u$ and information discriminator $D_I(x)$. This shows that the addition of the latent information variables not only helps control the appearance of the generated samples but also helps to increase the discriminability of the species.

The observed errors in the class discriminability test appear to mainly stem from class imbalance in the real training data, as artificial samples from species with more training samples generally achieve higher recognition accuracies. This issue might be reduced by balancing the distribution between the classes in the training. Additionally, the observed errors are also related to species, which are visually similar at the early growth stages. This is probably a result of the models incapability of reproducing small textural and shape details. Through visual inspection of the errors, it is also observed that some samples include attributes that are not present in the "true" species, but only in the "misclassified" samples. This shows attribute leakage between the species examples produced by the generator and that the generator has not fully learned what constitutes each species and what separates them. This leakage mainly occurs in the generated samples of larger plants. Many plant species develop their visual characteristics as they grow larger. However, in this work, we have removed many of the larger samples (>400 pixels in either dimension) to maintain a relatively small resizing factor in the preprocessing of the images. By including these larger images in the training, it might be possible to learn more about the individual species characteristics and avoid some of the above-mentioned errors. However, to include these larger images, we would need larger network architectures for the generator and the discriminator, which would make the training more unstable [6]. A potential solution could be to apply a more advanced conditioning scheme [21,22] or a progressive growing GAN training approach [29].

Section 3.2 showed that it is possible to remove samples with poor class representation from the artificial data distribution by evaluating the discriminator's confidence in each artificial sample. This enables us to only select samples that have a relatively higher resemblance to real plant seedlings, which is relevant if the samples are used for data augmentation purposes or in transfer learning, as previously shown in [13]. However, the selection process may remove some variability from the artificial samples, but probably not anything too severe since 86% of the samples remain even with a $D_C$ confidence threshold of >0.95.

## 5. Conclusions

In this paper, we have shown that the WacGAN-info model is capable of producing artificial samples from nine different species. The artificial samples resemble the intended species with an average recognition accuracy of 64.3% evaluated using an external state-of-the-art classification model, which is an improvement of 5.4% points compared to previous work. The observed errors are mostly related to class imbalances in the training data or to the applied network models, which are incapable pf reproducing small textural and shape details in the samples. We have also shown that it is possible to apply an unsupervised conditioning scheme in the training of WacGAN-info, which enables control over the visual appearance of the samples through a set of latent variables. Specifically, we show that it is possible to control the orientation and size of the samples through two latent variables. Finally, we have shown that it is possible to apply the discriminator as a quality assessment tool, which can be used to remove samples with poor class representation from the artificial data distribution, further increasing the recognition accuracy to 65.3%.

**Author Contributions:** Conceptualization, S.L.M. and A.K.M.; Data curation, S.L.M. and A.K.M.; Formal analysis, S.L.M. and A.K.M.; Funding acquisition, R.N.J.; Investigation, S.L.M. and A.K.M.; Methodology, S.L.M. and A.K.M.; Project administration, S.L.M.; Resources, R.N.J. and H.K.; Software, S.L.M. and A.K.M.; Supervision, R.N.J. and H.K.; Validation, S.L.M. and A.K.M.; Visualization, S.L.M. and A.K.M.; Writing—original draft, S.L.M. and A.K.M.; Writing—review and editing, S.L.M., A.K.M., R.N.J., and H.K.

**Funding:** This research was founded by Innovation Fund Denmark as part of the RoboWeedMaPS project (grant number 6150-00027B).

**Conflicts of Interest:** The authors declare no conflict of interest.

## Abbreviations

The following abbreviations are used in this manuscript:

| | |
|---|---|
| GAN | Generative adversarial network |
| WGAN | Wasserstein GAN |
| WGAN-GP | Wasserstein GAN w. gradient penalty |
| infoGAN | Information maximizing GAN |
| ACGAN | Auxiliary classifier GAN |
| $D$ | Discriminator network |
| $G$ | Generator network |
| EPPO | European and Mediterranean Plant Protection Organization |
| sPSD | Segmented plant seedling dataset |
| MDPI | Multidisciplinary Digital Publishing Institute |

## Appendix A. Implementation Details

### Appendix A.1. WacGAN-Info Model

Tables A1 and A2 provide implementation details on the discriminator and generator network architectures, respectively. The discriminator described in Table A1 is implemented as a convolutional neural network that inputs $128 \times 128$ RGB images and outputs three vectors: an estimate of the source distribution (fs-s), an estimate of the class (fc-c), and an estimate of the latent info variable (fs-i). The generator described in Table A2 is implemented as a transpose convolutional neural network that maps a vector to a $128 \times 128$ RGB image.

Table A3 provides a summary of the hyperparameters and other training settings used in the implementation of the GAN model.

**Table A1.** Discriminator network architecture details.

| | | | Discriminator $D(x)$ | | | | |
|---|---|---|---|---|---|---|---|
| Layer | Kernel Size | Stride | Output Shape | Activation Function | Batch Normalization | Dropout | Padding Scheme |
| input | - | - | $batch \times 128 \times 128 \times 3$ | - | - | - | - |
| conv1 | $[3 \times 3]$ | $[2 \times 2]$ | $batch \times 64 \times 64 \times 16$ | Leaky ReLU | No | 0.5 | Same |
| conv2 | $[3 \times 3]$ | $[1 \times 1]$ | $batch \times 62 \times 62 \times 32$ | Leaky ReLU | No | 0.5 | Valid |
| conv3 | $[3 \times 3]$ | $[2 \times 2]$ | $batch \times 31 \times 31 \times 64$ | Leaky ReLU | No | 0.5 | Same |
| conv4 | $[3 \times 3]$ | $[1 \times 1]$ | $batch \times 29 \times 29 \times 128$ | Leaky ReLU | No | 0.5 | Valid |
| conv5 | $[3 \times 3]$ | $[2 \times 2]$ | $batch \times 15 \times 15 \times 256$ | Leaky ReLU | No | 0.5 | Same |
| conv6 | $[3 \times 3]$ | $[1 \times 1]$ | $batch \times 13 \times 13 \times 512$ | Leaky ReLU | No | 0.5 | Valid |
| reshape | - | - | $batch \times 86528$ | - | - | - | - |
| fc-s | $[86528 \times 1]$ | - | $batch \times 1$ | None | No | 0.0 | - |
| fc-c | $[86528 \times N_C]$ | - | $batch \times N_C$ | Softmax | No | 0.0 | - |
| fc-i | $[86528 \times N_I]$ | - | $batch \times N_I$ | None | No | 0.0 | - |

**Table A2.** Generator network architecture details.

| | | | Generator $G(z, c)$ | | | |
|---|---|---|---|---|---|---|
| Layer | Kernel Size | Stride | Output Shape | Activation Function | Batch Normalization | Padding Scheme |
| input | - | - | $batch \times (128 + N_C + N_I)$ | - | - | - |
| fc | $[137 \times 768]$ | - | $batch \times 768$ | None | No | - |
| reshape | - | - | $batch \times 1 \times 1 \times 768$ | - | - | - |
| tconv1 | $[5 \times 5]$ | $[2 \times 2]$ | $batch \times 5 \times 5 \times 384$ | ReLU | Yes | Valid |
| tconv2 | $[5 \times 5]$ | $[2 \times 2]$ | $batch \times 13 \times 13 \times 256$ | ReLU | Yes | Valid |
| tconv3 | $[5 \times 5]$ | $[2 \times 2]$ | $batch \times 29 \times 29 \times 192$ | ReLU | Yes | Valid |
| tconv4 | $[5 \times 5]$ | $[2 \times 2]$ | $batch \times 61 \times 61 \times 64$ | ReLU | Yes | Valid |
| tconv5 | $[8 \times 8]$ | $[2 \times 2]$ | $batch \times 128 \times 128 \times 3$ | Tanh | No | Valid |

**Table A3.** WacGAN-info hyperparameters and training settings.

| Parameter | Value |
|---|---|
| Unstructured noise dimension | 128 |
| Batch size | 64 |
| Class scaling ($w_C$) | 7.5 |
| Information scaling ($w_I$) | 15 |
| Training ratio ($D : G$) | 5:1 |
| Gradient penalty coefficient ($\lambda$) | 10 |
| Leaky ReLU slope | 0.2 |
| Optimizer (Discriminator) | Adam ($lr = 0.0002, \beta_1 = 0.5, \beta_2 = 0.9$) |
| Optimizer (Generator) | Adam ($lr = 0.0010, \beta_1 = 0.5, \beta_2 = 0.9$) |
| Weight, bias initialization | xavier, constant(0) |
| Training iterations | 12,000 epochs |

*Appendix A.2. External ResNet-101 Classifier*

TensorFlow-Slim [30] implementation of ResNet-101 [25] was used as the base network for the external classifiers. A ResNet-101 model pretrained on ImageNet [17] provided by TensorFlow-Slim was used for initialization of all the network weights. The hyperparameters used for training the network are summarized in Table A4. Training images were augmented in four steps. First, they were zero-padded to $566 \times 566$ pixels, a random rotation was applied, they were cropped to $400 \times 400$ pixels, and finally, they were resized to $128 \times 128$ pixels to match the input size of the WacGAN-info models. During evaluation, the images were subject to the same augmentation, except for the random rotation.

**Table A4.** ResNet101 hyperparameters and training settings

| Parameter | Value |
|---|---|
| Batch size | 24 |
| Optimizer | Stochastic gradient descent ($lr = 0.001$) |
| Weight, bias initialization | Pre-trained model on ImageNet |
| Training iterations | 500 epochs |

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
