# Peer review of "Disentangling Information in Artificial Images of Plant Seedlings Using Semi-Supervised GAN"

_remotesensing, doi:10.3390/rs11222671_

Round 1

Reviewer 1 Report

This manuscript presents a DNN to generate artificial samples of plant images. This work builds upon the authors' previous work [9], improving the network by adding info-gan and wgan architectural and training designs.

Overall, I am not exciting by this paper as I dont see any novelty. I will explain more in detail:

The introduction doesnt show what is novel with this paper wrt with their previous work and the current literature. For what I can see, nothing is novel there The way the network is explained is fuzzy. For example, the meaning of z,y, and u come very late, compared with the reference to fig 1 the authors do at page 2. The structure is not optimal and it is hard to read  Fig 2 shows a big panel of images and claim those images are visually good (l 172). They also compare with their previous work. However, there is no image comparing images produced by this work, their previous work, and the original dataset. For what I can see, these images and the ones in [9] look the same. Moreover, the images appear textureless and, as they say several times, they have boundary artifact. These artifacts were present also in their previous work. So my question is, why the authors did not focus in the image quality, before to go to a more complicate learning method and architecture, which allow them to achieve the same kind of images of their previous work? The experiment generating fig 6 is not clear and I failed to see what it means. In line 260, the authors say that the classifier is able to classify the plant species with high degree. 64% (+5% improvement from their previous work) is not something extremely exceptional. The authors blame the class imbalance for some of their bad results. Since they had their previous work in [9] to generate artificial images, why didnt they use it to solve the class imbalance problem? Maybe because, at the end of the day, the generated images are far away from the original images in the dataset? Some typos can be see in the paper. eg line 272 "an result".

Since the overall tone of the paper and the results show no excitement and novelty, I would reject this manuscript as it is now.

Reviewer 2 Report

The authors propose a novel Generative Adversarial Network (GAN), namely WGAN-info which combines WGAN and infoGAN for generating artificial plant images. Using such networks, the authors successfully generate synthetic plant images which the independently trained classifier displays the classification accuracy of 64.3%. They claim that such generated datasets can be utilized as a pre-annotated training data for a supervised machine learning.

Comments

The actual usage of GAN derived data as useful pre-annotated dataset for wide applications, which the authors claim, is not described nor included in the manuscript. As I leave it to the editor’s decision whether the actual application must be incorporated to demonstrate the authors claim to meet the journal policy, I feel at least some of the literatures if present, must be appropriately cited to reinforce. This is because whether the GAN generated data to augment a certain dataset is useful or not, is in still debate in various domain, where the cons claim that it is no more than an interpolation of the existing dataset, which is no more than an image augmentation. The authors should organize and emphasize more on the advantage and novelty of using WGAN-info compared to the previously reported WGAN model. In the current form, this manuscript can be interpreted as a “yet another GAN for generating plant images”, similar to what the author report recently (Madsen et al., 2019). The advantage can be in any metrics, the aid of semi-supervised, the output of the generated results, training efficiency, etc. Should be an independent paragraph or section to claim novelty. Similar to the above, the authors should display the visual output of the WGAN (Madsen et al., 2019) (as well as the actual plant classs images) suppose in Fig.2, since the comparison between the different methods are explained multiple times in the manuscript. This is also important related to the 2nd

Others

Should unify the abbreviation either to WacGAN or WGAN.

Reviewer 3 Report

This paper addresses one of the main problems encountered in plant research, where data are considerably reduced compared to other general object recognition tasks. The authors proposed using the GAN approach to extend the data and adding an unsupervised learning branch that resembles the characteristics of InfoGAN in their previously proposed method [1]. Overall, the paper structure is clear, and the quantitative and qualitative analysis showed that the author carried out a critical evaluation of the proposed WacGan-Info. Here are my few comments:

First of all, there is no comparison between WacGan-Info and standard GAN approaches such as InfoGan[2] and ACGAN[3], which makes me wonder what is missing in existing standard approaches that WacGan-Info should be proposed. I would say that it would be useful to improve the experimental part, at least in Sec 3.2 to compare the class discrimination of WacGan-Info with other standard GANs to see if the samples generated by WacGan-Info could infer a better result.

Then, on line 126-131, I don't understand why you explain that you don't use the Inception score [4] as it is trained on other application but rather use ResNet-101. If a CNN model is to be used in the field of plant seedlings, either one of them is also capable of fine-tuning, isn’t it?

Finally, on line 167, I think you are talking about WacGan-Info.

[1] Madsen, Simon L., et al. "Generating artificial images of plant seedlings using generative adversarial networks." Biosystems Engineering 187 (2019): 147-159.

[2] Chen, Xi, et al. "Infogan: Interpretable representation learning by information maximizing generative adversarial nets." Advances in neural information processing systems. 2016.

[3] Odena, Augustus, Christopher Olah, and Jonathon Shlens. "Conditional image synthesis with auxiliary classifier gans." Proceedings of the 34th International Conference on Machine Learning-Volume 70. JMLR. org, 2017.

[4] Szegedy, Christian, et al. "Rethinking the inception architecture for computer vision." Proceedings of the IEEE conference on computer vision and pattern recognition. 2016.

Round 2

Reviewer 1 Report

The authors have edited the manuscript and provided a response letter to my comments. In general, for what I can see, the changes the authors made are minimal for a paper that has got a reject score. I will further augment my thoughts and how they were dealt by the authors:

The authors stated that the primary focus of the paper is to get a better control over the plant visual appearance. Does it happens? In some extends it does, but it is not perfect. The visual appearance the authors mentioned is, according to the Fig4, only about rotation (not extremely useful for plants), and size (this is useful). However, does not give control on more important biological details. In [11] and [12] the authors have control on one very important features of plants: the number of leaves. But they deal with that in a supervised fashion. Having this aspect in an unsupervised fashion would have been more useful for plant biologists.

The appendix the authors added are panels obtained from their previous work. What I meant is to have  one figure where original images, WacGAN and WacGAN-info images come together and it would have been easier to visually inspect those. In this case instead, I need to do weird stuff to compare images and I could not understand whether the images from the 'new proposed model' are better than the previous one. This important comparison comment was poorly addressed  

The authors also state in the response letter that the increase of quality is shown in manuscript. I have probably missed that because I still believe the quality of the image is below compared to the original images in the appendix A1.

For the above reasons, I wouldnt recommend this paper for publications.

I understand the authors did not want to augment the dataset to compare the results with their previous work. However, noone prevented them to try it and train a new model to see if it would increase the accuracy/image quality. It's ok to compare with other work, but further analysis could also have been conducted and show more information to support the goodness of their images. At the end of the day, why do you need to generate synthetic images? The reason I care is to better train algorithms when we have lack of data. This important test was not performed and I cannot see any other applications to generate synthetic images than this.

Reviewer 2 Report

The authors have honestly showed their efforts to resolve my previous comments and or concerns. With additional graphical figures, I think the paper is more attracting to the readerships by contrasting a superior output of wacGAN-info compared to the previous methods.

Author Response

Dear Reviewer

Thank you for your valuable feedback on our manuscript named: “Disentangling information in artificial images of plant seedlings using semi-supervised GAN”.

We appreciate that you find our work suitable for publication and that you have taken your time to review our manuscript.

We made a minor revision of the manuscript based on feedback from another reviewer. The figures visualizing samples from WacGAN-info (previously, figure 2), sPSD (previously, Figure A1), and WacGAN (previously, Figure A2) have been combined into a single table (Table 2). This new table includes samples from all three sources, which should make visual comparisons between the samples easier.

Reviewer 3 Report

My initial comments have been sufficiently addressed in the revised manuscript.

Author Response

(The authors gave the same response as above.)
